# Metformin lotion promotes scarless skin tissue formation through AMPK activation, TGF-β1 inhibition, and reduced myofibroblast numbers

Jianying Zhang[1], Kengo Shimozaki[1], Soichi Hattori[1], Vasyl Pastukh[1], Derek Maloney[1], MaCalus V. Hogan[1,2], James H-C. Wang[1,2,3]*

1 MechanoBiology Laboratory, Department of Orthopaedic Surgery, University of Pittsburgh, Pittsburgh, PA, United States of America, 2 Department of Bioengineering, University of Pittsburgh, Pittsburgh, PA, United States of America, 3 Department of Physical Medicine and Rehabilitation, University of Pittsburgh, Pittsburgh, PA, United States of America

* wanghc@pitt.edu

## Abstract

Scar tissue formation following skin wound healing is a challenging public health problem. Skin regeneration and preventing the formation of scar tissue by currently available commercial products are largely ineffective. This study aimed to test the efficacy of a novel topical metformin lotion (ML) in inhibiting scar tissue formation during skin wound healing in rats and to determine the mechanisms of action involved. A 6% ML was prepared in our laboratory. A skin wound healing model in rats was used. The wounded rats were divided into two groups and treated daily for 10 days as follows: Group 1 received a daily application of 50 mg of control lotion, or 0% ML (totaling 100 mg of lotion per rat), and Group 2 received a daily application of 50 mg of 6% ML (totaling 100 mg of 6% ML per rat). Blood samples from the heart of each rat were analyzed for inflammatory markers, HMGB1 and IL-1β, using ELISA, and immunological and histological analyses were performed on skin tissue sections. ML decreased levels of inflammatory markers HMGB1 and IL-1β in the serum of rats and inhibited the release of HMGB1 from cell nuclei into the skin tissue matrix. Additionally, ML demonstrated anti-fibrotic properties by enhancing AMPK activity, decreasing the expression of TGF-β1, reducing the number of myofibroblasts, decreasing the production of collagen III, and increasing the expression of collagen I. ML promotes the regeneration of high-quality skin during wound healing by reducing scar tissue formation. This effect is mediated through the activation of AMPK, inhibition of TGF-β1, and a decrease in the number of myofibroblasts.

## Introduction

Skin wounds and impaired wound healing pose significant challenges to public health. In the United States, chronic wounds affect 6.5 million patients, with annual healthcare costs

**Data Availability Statement:** Data will be available in the public repository after acceptance. URL and

accession numbers will be available after acceptance.

**Funding:** This work was supported in part by a DOD/MTEC Award (W81XWH2290016) https://mtec-sc.org/ to JW and Pittsburgh Foundation Albert B. Ferguson, Jr. MD. Orthopaedic Fund Awards (AD2021-120112; AD2022-130408; AD2023-134256) to JZ. The funders did not play any role in the study design, data collection and analysis, decision to publish, or preparation of the manuscript.

amounting to approximately \$25 billion [1]. The natural process of skin recovery following an injury typically results in scar formation. Scars, often resulting from burns or surgical procedures, significantly impact both individuals and society at large [2]. Each year in the United States, about 500,000 individuals receive treatment for burns, many of which result in scarring and painful contractures necessitating extensive surgical intervention [3]. Additionally, around 100 million patients in developed countries develop surgical scars annually [4].

Scars can be painful, disfiguring, and disabling, leading to significant distress. Patients with visible scars, particularly on the face, often face social stigma and experience psychological trauma [4, 5]. In the United States, the market for scar treatment is estimated at \$12 billion annually. Despite this, many commercially available wound healing products are ineffective at promoting skin regeneration and preventing the formation of scar tissue [6].

Wound healing is a complex process that includes stages such as blood clotting, inflammation, cellular proliferation, and remodeling of the extracellular matrix (ECM) [7]. The inflammatory response that occurs after tissue injury is critical for both normal and pathological healing processes [8]. It has been noted that inflammation significantly influences the later stages of tissue repair, which may lead to scarring [9]. High Mobility Group Box 1 (HMGB1), found in almost all cells and a potent mediator of inflammation, is known to play a significant role in the pathology of various inflammatory diseases, particularly in wound healing [10]. HMGB1's extracellular actions vary depending on the redox state of its cysteine residues, which can either promote healing or exacerbate inflammation [11]. Fully reduced HMGB1 (frHMGB1) aids in recruiting cells like leukocytes and stem cells, thereby supporting the regeneration of damaged tissues [12]. Conversely, partially oxidized disulfide HMGB1 (dsHMGB1) triggers leukocyte activation and the release of pro-inflammatory cytokines, which can be detrimental to tissue healing [13].

Recent research has identified the signaling molecule HMGB1 as a critical factor in the fibrotic changes observed in systemic sclerosis, as well as liver, kidney, and lung fibrosis [14–16]. This protein, released following tissue damage, appears to facilitate wound healing by attracting cells to the injury site. However, an excessive accumulation of these cells can lead to increased scar tissue formation. Interestingly, inhibiting HMGB1 has been found to significantly reduce scarring [17, 18]. This highlights HMGB1's dual role in scar management and suggests its potential as a therapeutic target in treating fibrotic diseases.

Scar tissue formation is marked by an excessive buildup of ECM components, primarily produced by myofibroblasts that are identified by their expression of alpha-smooth muscle actin ($\alpha$-SMA) [19]. Elevated levels of $\alpha$-SMA and collagen III lead to the development of loose collagen fibers, which are instrumental in scar formation. Adenosine monophosphate-activated protein kinase (AMPK) plays a crucial role in regulating inflammatory signals. The activation of AMPK has been shown to reduce scar tissue formation [20].

Metformin, a drug widely used to manage diabetes, possesses anti-oxidant and anti-scarring properties [21–23]. In particular, metformin exhibits significant anti-inflammatory properties by inhibiting the production of prostaglandin$E_2$ ($PGE_2$) and proinflammatory cytokines such as interleukin-1 $\beta$ (IL-1$\beta$), interleukin-6 (IL-6), and tumor necrosis factor-$\alpha$ (TNF-$\alpha$). Metformin can mitigate oxidative stress and support tissue healing. Metformin interacts with the HMGB1, specifically binding to its C-tail and inhibiting its activity [24]. This interaction has been linked to metformin's capacity to counteract fibrosis in several organs including the lungs [25–27] and joint capsules [28]. Further investigations reveal that metformin enhances wound healing by altering the behavior of fibroblasts and macrophages within the wound microenvironment, activating AMPK, and decreasing levels of the fibrogenic cytokine, transforming growth factor- $\beta$1 (TGF-$\beta$1) [29]. These findings underscore metformin potential beyond glucose regulation, particularly in its anti-fibrotic effects across multiple tissue types.

However, metformin is typically administered orally, which can lead to side effects in the stomach, liver, and kidneys, and it is less effective for skin wounds. In this study, we developed a metformin-based lotion designed for topical application to directly target skin wounds and enhance healing. This topical formulation of metformin increases its efficacy by requiring smaller doses than oral administration, thereby minimizing systemic side effects. The effectiveness of this ML on skin wound healing was evaluated using a rat model.

## Materials and methods

### Animals

The animal study protocols received approval from the University of Pittsburgh's Institutional Animal Care and Use Committee (IACUC), under protocol numbers 22091916 and 22081551. All procedures involving animal usage, handling, surgery, anesthesia, analgesia, and euthanasia were conducted in strict accordance with established guidelines and regulations.

### Materials

Metformin hydrochloride (Cat. #PHR1084, pharmaceutical secondary standard), glycerol (Cat. #15524, meets analytical specification of Ph.Eur., BP), Vaseline (Cat. #16415, meets analytical specification of Ph. Eur), paraffin oil (Cat. #18512, meets analytical specification of Ph. Eur., BP), and sorbitan sesquioleate (Span 83; Cat. #S3386) were obtained from Millipore Sigma (Burlington, MA).

### Metformin lotion preparation

The metformin lotion was prepared under optimized conditions with a proprietary process. Our extensive testing confirmed that the lotion spreads uniformly across the skin, facilitating the efficient diffusion of metformin through the dermal layers. Note that Met lotion is a water-based product with a higher concentration of metformin, lower viscosity, and better skin penetration compared to creams, ointments, and gels, as metformin is a water-soluble compound.

### Animal numbers and skin wound healing model

Our study consists of two groups: the vehicle control group and the 6% ML group. We estimated the effect size of Met treatment on HMGB1 levels to be at least 1.8 [30]. In addition, the following parameters were used to calculate animal numbers using G-Power: $\alpha = 0.05$, Power $(1 - \beta) = 0.8$, and allocation ratio (N2/N1) = 1. The resulting sample size is 10. Hence, ten 3-month-old female Sprague Dawley (SD) rats, with an average weight of 200 g, were used in this study.

The animals were anesthetized with isoflurane (2%-3%), and the procedure was performed under aseptic conditions, utilizing sterile gloves, masks, sterile instruments, and aseptic techniques. Under sterile condition, the hairs were removed from the skin over the Achilles tendon area by a shaver, sprayed with 70% ethanol and swabbed with iodine. A 2 cm incised wound was carefully created on the skin over each Achilles tendon area using a scalpel, after which the wounds were sutured. Subsequently, the wounded rats were divided into two groups and treated daily for 10 days as follows. **Group 1:** Each leg's skin surface received a daily application of 50 mg of 0% ML (vehicle lotion, totaling 100 mg of 0% ML per rat). **Group 2:** Each leg's skin surface was treated daily with 50 mg of 6% ML (totaling 100 mg of 6% ML per rat) (**Fig 1A–1D**). To alleviate post-surgical pain, the animals received carprofen (5 mg/kg body weight) during the surgery and twice daily for four days following the procedure.

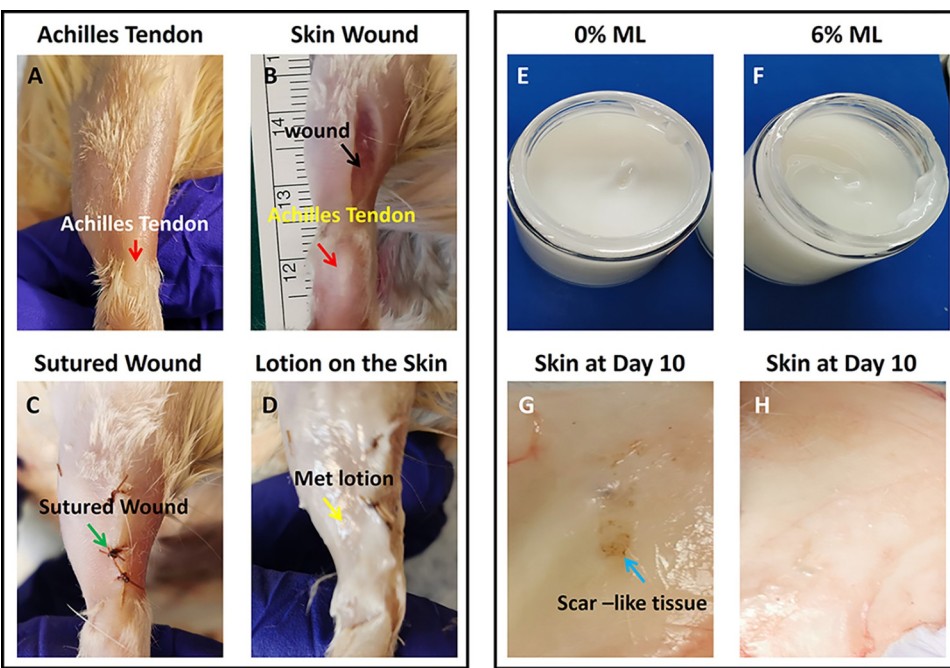

**Fig 1. Photographs show the appearance of ML and rat skin over the Achilles tendon before and after surgery. A:** The rat skin over the Achilles tendon (red arrow labeled area) was prepared for surgery after removing the hair by a shaver. **B:** A wound (2 cm) was created in the skin over the Achilles tendon (black arrow labeled area). **C:** The skin wound was sutured (green arrow labeled area). **D:** The ML was applied on the sutured skin wound area (yellow arrow labeled material). **E:** The appearance of the 0% ML shows a white cream-like product. **F:** The appearance of the 6% ML shows a white cream-like product. **G:** Gross inspection shows some scar-like tissue appeared in the inside of the skin over the Achilles tendon in 0%ML. **H:** Gross inspection shows some scar-like tissue appeared in the inside of the skin over the Achilles tendon in 6%ML.

Ten days post-surgery, the rats were humanely sacrificed using $CO_2$ asphyxiation, followed by thoracotomy to confirm death. Blood samples were then collected from the heart of each rat and used for inflammation testing via enzyme-linked immunosorbent assay (ELISA). Additionally, skin tissues from the Achilles tendon area were harvested for further analysis. The effect of metformin on wound healing was evaluated through histological examination of the skin tissue sections.

## Measurement of HMGB1 and IL-1β in the serum of rats by ELISA

The HMGB1 and IL-1β concentrations in rat serum samples were measured using respective ELISA kits (Cat. #NBP2-62767; Novus Biologicals; Centennial, CO, USA; Cat. #ab255730; Abcam; Boston, MA, USA) following the manufacturer's instructions. Blood samples were obtained from the rats' hearts and left at room temperature for 1 hour before being centrifuged at $1000 \times g$ for 15 minutes. After centrifugation, the supernatant was carefully separated from the red blood cell pellets. If the HMGB1 and IL-1β levels were not immediately assayed, the supernatant was stored at -20°C for later analysis.

## Histochemical staining for the structural analysis of rat skin tissues

Skin tissues from the Achilles tendon areas of the rat hind legs were carefully dissected and then immersed in 4% paraformaldehyde overnight at 4°C for fixation. The fixed skin tissues were embedded in paraffin and sliced into 5 μm thick sections. These tissue sections were

examined for structural analysis with hematoxylin and eosin (H&E), and Masson's trichrome (MT) staining, according to the standard protocols. The stained tissue sections were subsequently examined under a light microscope (Nikon eclipse, TE2000-U).

### Picro Sirius red staining and polarized light microscopy of rat skin tissue sections

The paraffin-embedded rat skin tissue block was sectioned into slices 5 μm thick and stained using a Picro Sirius Red kit (Cat. #ab150681, Abcam, Waltham, MA, USA) according to the manufacturer's instructions. Subsequently, the stained skin tissue sections were analyzed under a polarized light microscope (Nikon).

### Immuno-fluorescent staining on rat skin tissue sections

For immunofluorescent staining, the fixed rat skin tissue samples were promptly embedded in O.C.T compound (Sakura Finetek USA Inc., Torrance, CA) within disposable molds and then frozen at -80°C. Subsequently, cryostat sectioning was carried out at -25°C to obtain approximately 5 μm thick tissue sections, which were left to equilibrate to room temperature overnight. Following this, the tissue sections underwent fixation in 4% paraformaldehyde for 15 minutes, followed by three washes with PBS.

For staining HMGB1 and AMPK, the fixed tissue sections were treated with 0.1% Triton-X-100 at room temperature for 30 minutes, then rinsed with PBS three times. The treated tissue sections were then incubated overnight at 4°C with either rabbit anti-HMGB1 antibody (1:330; Cat. #ab18256, Abcam; Waltham, MA, USA), rabbit anti-AMPK antibody (1:500; Cat.

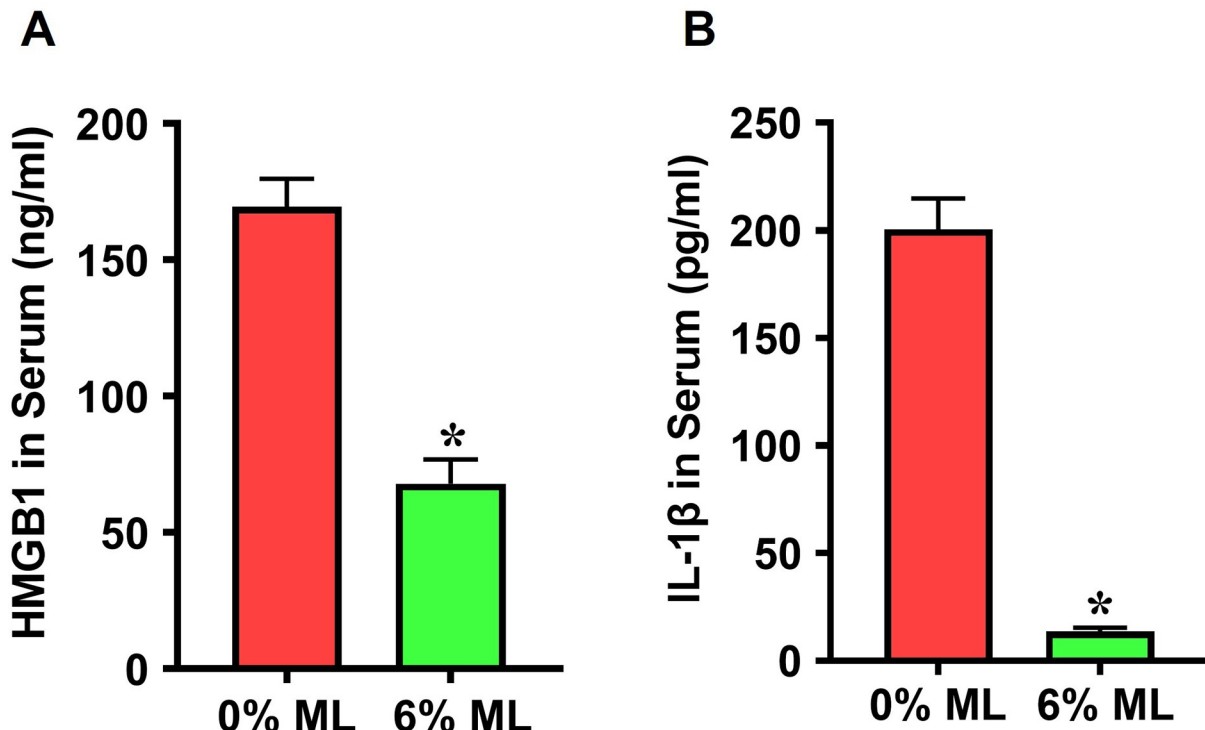

**Fig 2. Metformin lotion is anti-inflammatory in serum following skin wounds.** The levels of HMGB1 (**A**) and IL-1β (**B**) in the serum of the rats with skin wounds treated with 6% ML for 10 days are significantly lower compared to 0% ML treated group. ML: Metformin Lotion. *p < 0.05, n = 3–5. Analysis was conducted by ELISA.

#MA5-15815, ThermoFisher Scientific; Waltham, MA), rabbit anti-phospho-AMPK antibody (1:500, Cat. #ab133448, Abcam, Waltham, MA), rabbit anti-α-SMA antibody (1:500; Cat. #ab124964, Abcam, Waltham, MA, USA), rabbit anti-TGF-β1 antibody (1:500; Cat. #ab215715, Abcam; Waltham, MA, USA), rabbit anti-collagen I antibody (1:500; Cat. #ab138492, Abcam, Waltham, MA, USA), or rabbit anti-collagen III antibody (1:500; Cat. 184993, Abcam, Waltham, MA, USA).

The following morning, the tissue sections were rinsed five times with PBS and then incubated with goat anti-rabbit secondary antibody conjugated with Cy3 (1:500, Cat. #AP132C; Millipore, Burlington, MA) for 2 hours at room temperature. Finally, the tissue sections were counterstained with DAPI, and the positively stained results were examined using a fluorescent microscope (Nikon, Eclipse TE2000U).

## Semi-quantitative assessment of positive stained cells in rat skin tissue sections

Three random images were selected from each skin tissue section using a Nikon Eclipse TE2000-U microscope to generate semi-quantitative staining results. This process yielded a total of nine images analyzed for each group, reflecting three sections from three different rats.

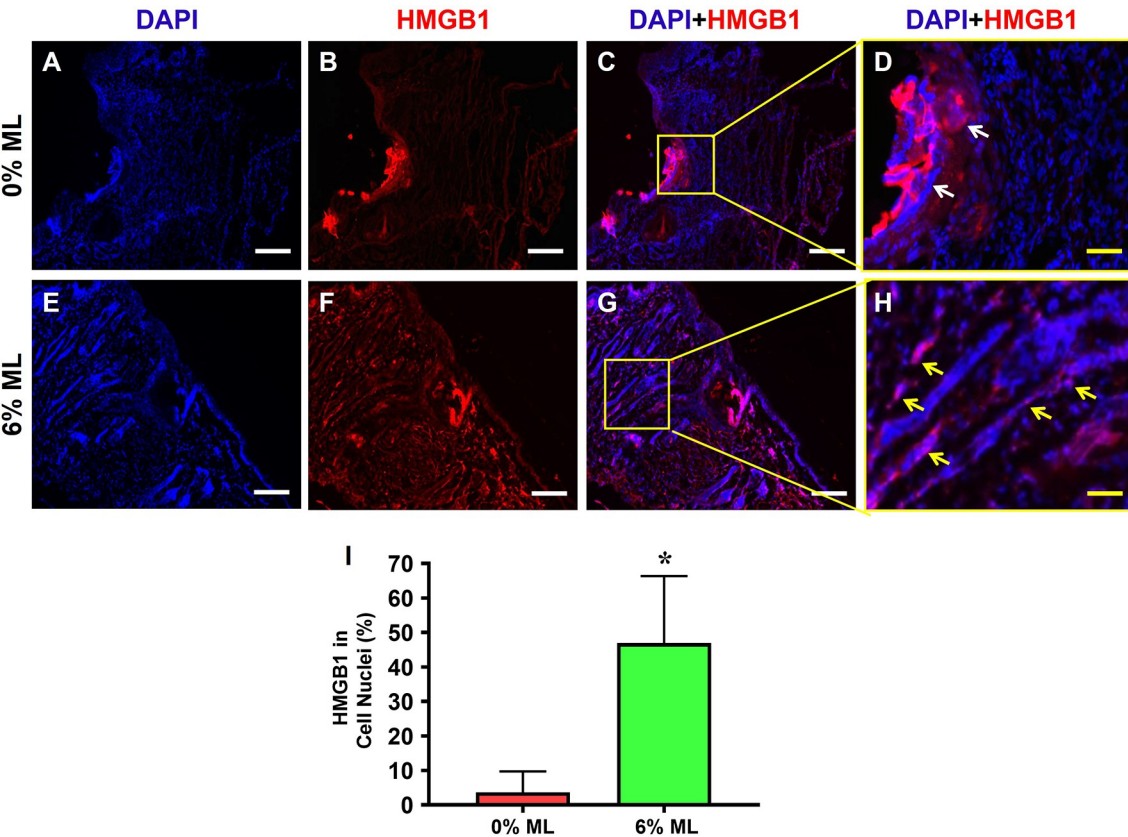

**Fig 3. Metformin lotion inhibits the release of HMGB1 from cell nuclei to the skin tissue matrix in rats.** High levels of HMGB1 are present in the tissue matrix of rats treated with 0% ML (white arrows in **D**). However, in the 6% ML treated group (**E-H**), HMGB1 is predominantly localized within the cell nuclei (yellow arrows in **H**). Semi-quantification results agree with this finding (**I**). **C**: Merged images of **A** and **B**. **G:** Merged images of **E** and **F**. **D**: Enlarged yellow box area in image **C**. **H**: Enlarged yellow box area in image **G**. ML: Metformin Lotion. *p < 0.001, compared to 0% ML treated wounds. Scale bars: 200 μm (white), 50 μm (yellow). Analysis was conducted by immunostaining.

Positive staining cells within the tissue sections were manually identified by reviewing the captured images and subsequently analyzed using SPOT imaging software (Diagnostic Instruments). The percentage of positive staining was calculated using the formula: % of positive staining = (number of positively stained cells / total number of cells stained with DAPI) × 100%. The mean percentage of positive staining for each group was then determined by averaging the resulting values.

## Statistical analysis

The data were analyzed using one-way ANOVA followed by Fisher's PLSD for multiple comparisons. A p-value of less than 0.05 was considered to indicate a significant difference between two groups.

## Results

We developed a metformin-based lotion for the treatment of rat skin wounds. This study compared the effects of a control lotion (0% ML) and a 6% ML, both of which are presented as white, smooth creams (**Fig 1E and 1F**). Gross inspection showed some scar-like tissues on the inside of the skin treated with 0% ML after 10 days healing (**Fig 1G**). However, normal-like tissues were found in the inside of the skin treated with 6% ML after 10 days healing (**Fig 1H**).

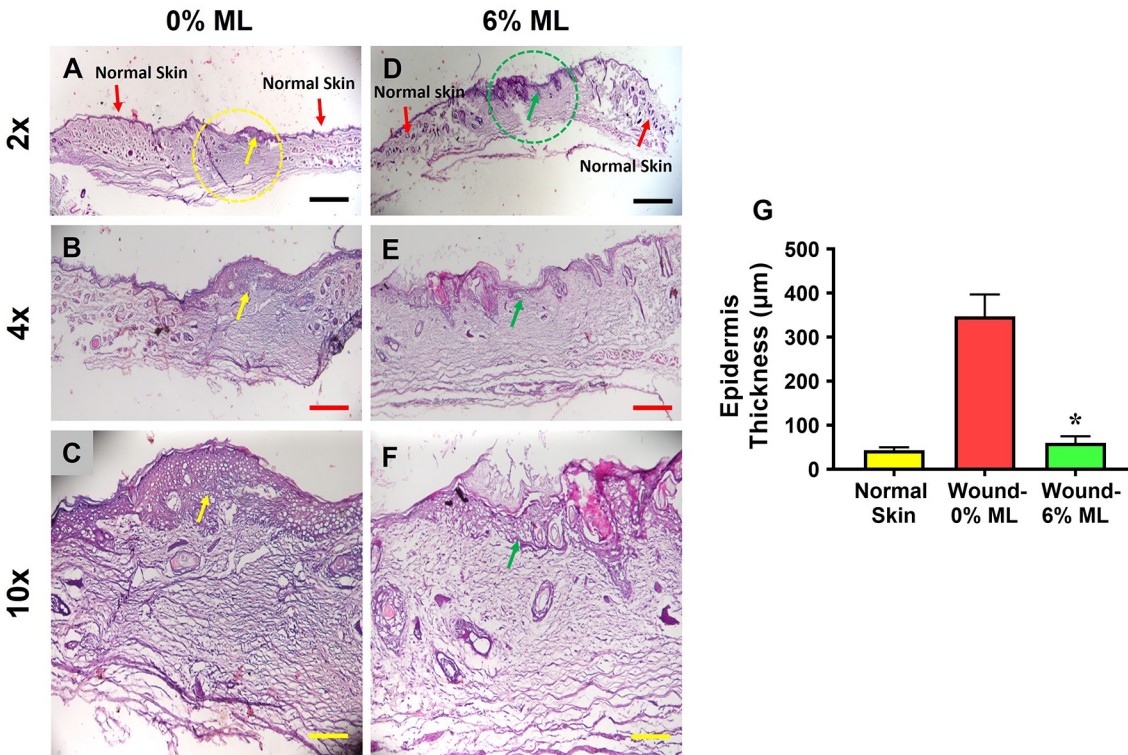

**Fig 4. Metformin lotion inhibits scar tissue formation in rat skin after 10 days of wound healing.** The results show a dense and thick epidermis formed at the wound area treated with 0% ML (yellow dashed line and arrows in **A-C**). In contrast, normal skin-like tissues formed at the wound area treated with 6% ML (green dashed line and arrows in **D-F**). Semi-quantification indicates that the epidermis thickness at the wound area treated with 0% ML is six times greater than at the wound area treated with 6% ML (**G**). There is no significant difference in epidermis thickness between normal skin and 6% ML-treated wounded skin. ML: Metformin Lotion. *$p < 0.05$ compared to 0% ML treated wounds. Scale bars: 1 mm (black), 500 μm (red), and 200 μm (yellow). Analysis was conducted using H&E staining.

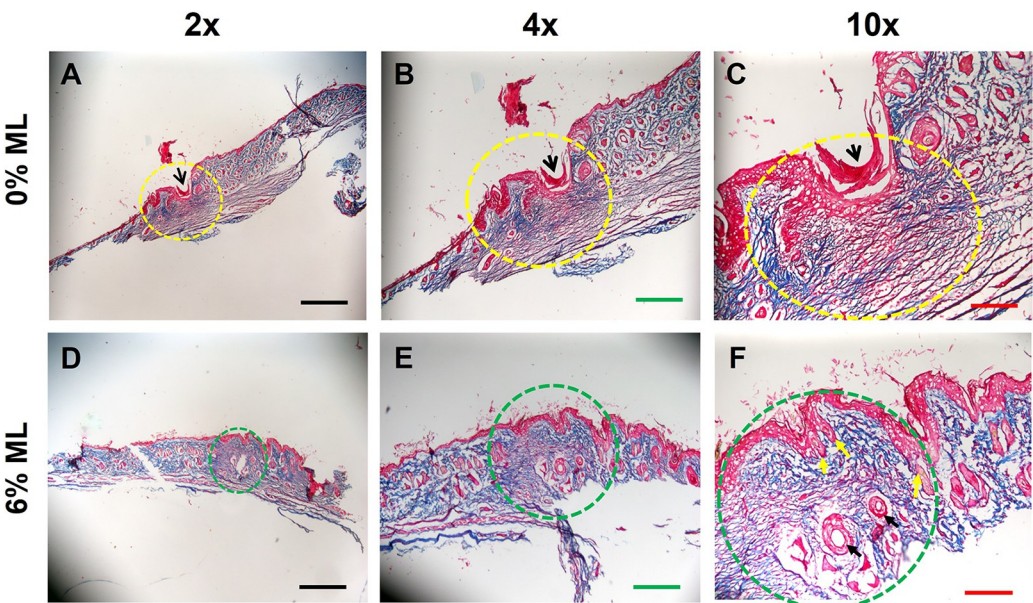

| 2x | 4x | 10x |

**Fig 5. Metformin lotion inhibits scar tissue formation in rat skin after 10 days of wound healing.** The results show a large gap area (black arrows in **A-C**) found at the wound area treated with 0% ML (yellow dash line area in **A-C**). However, no gap is found at the wound area treated with 6% ML (green dash line area in **D-F**). Some normal skin-like tissues with hair follicles (yellow arrows in **F**) and blood vessels (black arrows in **F**) form at the wound area treated with 6% ML (**D-F**). Few hair follicles are found in the wound area treated with 0% ML (**A-C**). Many fibrosis-like tissues are found in the 0% ML treated wound area (yellow dash line area in **A-C**). However, there are much less fibrosis-like tissues found in the 6% ML treated wound area (green dash line area in **D-F**) compared to 0% ML treated wound. ML: Metformin Lotion. Scale bars: 1 mm (black), 500 μm (green), and 200 μm (red). Analysis was conducted by MT staining.

Application of the 6% ML on wounded rat skin demonstrated an anti-inflammatory effect, as confirmed by ELISA assays. Specifically, serum levels of two key inflammatory markers, HMGB1 and IL-1β, were significantly reduced in rats treated with the 6% ML compared to those treated with the control lotion, or 0% ML (**Fig 2**). Specifically, HMGB1 levels decreased from 169.6 ng/ml in the control group, 0% ML, to 67.9 ng/ml following treatment with the 6% ML (**Fig 2A**). Similarly, IL-1β concentrations dropped from 200.5 pg/ml in the 0% ML group to 13.7 pg/ml in the 6% ML group (**Fig 2B**).

Immunostaining analyses of HMGB1 further supported these findings, showing higher levels of HMGB1 in the skin matrix surrounding the wound in control group (**Fig 3A–3D**). In contrast, wounds treated with 6% ML exhibited substantially lower HMGB1 levels in the surrounding skin matrix (**Fig 3E–3H**). Additionally, over 46% of cells in the wound areas of rats treated with 6% ML showed nuclear expression of HMGB1, significantly higher than the less than 3.7% observed in the control group (**Fig 3C, 3D and 3I**).

H&E staining results revealed that a dense and thick epidermis formed at the wound area treated with 0% ML (indicated by yellow dashed lines in **Fig 4A** and yellow arrows in **Fig 4A–4C**). In contrast, wounds treated with 6% ML developed normal skin-like tissues (marked by green dashed lines in **Fig 4D** and green arrows in **Fig 4D–4F**). Semi-quantitative analysis showed that the epidermis at the wound site treated with 6% ML was six times thinner compared to that treated with 0% ML, as shown in **Fig 4G**.

MT staining highlighted a large gap (black arrows in **Fig 5A–5C**) in the skin wound area treated with 0% ML (yellow dashed line area in **Fig 5A–5C**). The majority of tissues in the area circled with the yellow dotted line was stained blue, indicating a high level of collagen. This is typical of scar tissue, where collagen is laid down extensively to repair the damage. The red

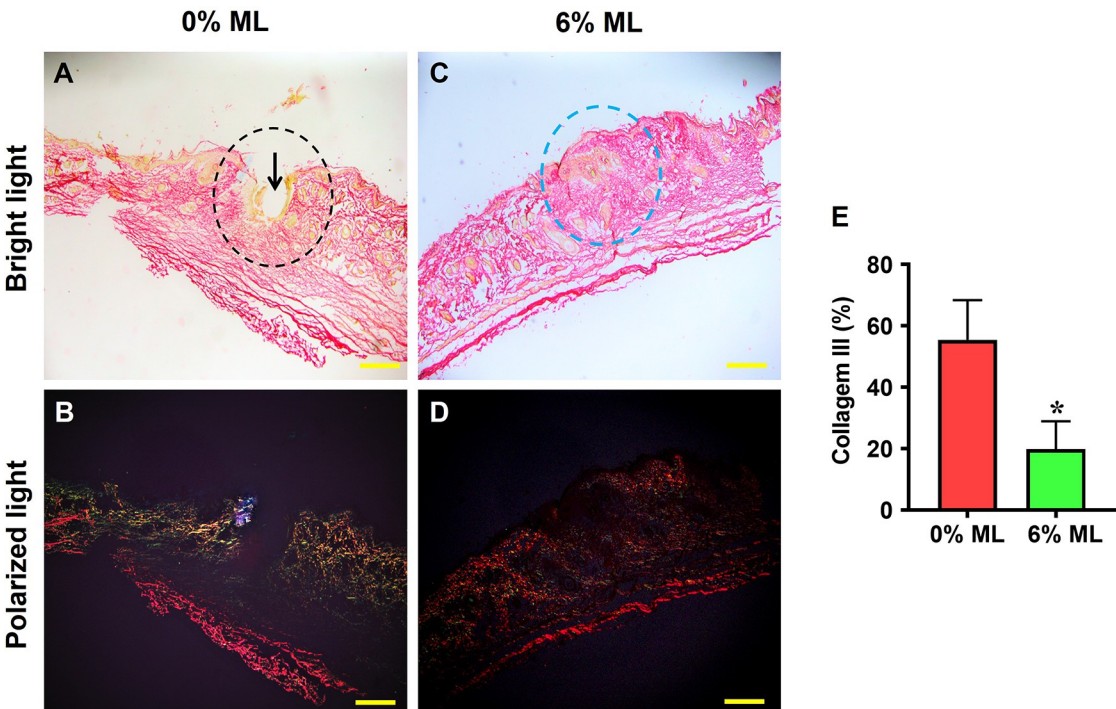

**Fig 6. Metformin lotion decreases collagen III levels in the wounded skin of rats after 10 days of wound healing.** Under a bright-field microscope, a large gap (black arrow in **A**) is found in the 0% ML treated wound areas (black dashed line in **A**), while no gap is found in the wound area treated with 6% ML (blue dashed line in **C**). Under polarized light microscopy, high levels of collagen III are found in the 0% ML treated wound area (green fluorescence in **B**), whereas the wound area treated with 6% ML is positively stained for collagen I (red fluorescence in **D**). Semi-quantification confirms these results (**E**). *p < 0.001, compared to the 0% ML treated skin wound. ML: Metformin Lotion. Scale bars: 500 μm (yellow). Analysis was conducted using Picro-Sirius Red staining.

staining, visible especially in areas like the one indicated by the black arrow, represents blood vessels. Overall, the structure of the tissue within the curved yellow line was somewhat less organized and denser compared to typical skin tissue, which is characteristic of scar formation where the normal architecture is disrupted and replaced by fibrotic tissue.

In contrast, the wound areas treated with 6% ML (green dashed line area in **Fig 5D–5F**) showed no gaps. The green dotted circle highlights an area of skin that includes both the epidermis (upper layers stained mostly red) and the underlying dermis (stained predominantly blue due to collagen presence). This clear layer delineation indicates good tissue regeneration. The collagen within this circled area appears well-organized and less dense compared to typical scar tissue, suggesting a more organized and functional dermal regeneration. Blood vessels (indicated by black arrows in **Fig 5F**) are clearly visible and well-defined with red staining, indicating adequate vascularization supporting the healing skin tissue. Overall, these areas developed normal skin-like tissues, including hair follicles (yellow arrows in **Fig 5F**). In comparison, very few hair follicles were present in the wound areas treated with 0% ML (**Fig 5A–5C**).

Picro Sirius Red staining under a bright-field microscope revealed a large gap in the wound areas treated with 0% ML (indicated by a black arrow in **Fig 6A**), while no gap was observed in the areas treated with 6% ML (marked by a blue dashed line in **Fig 6C**). Under polarized light microscopy, high levels of collagen III were detected in the wound areas treated with 0% ML (shown as green fluorescence in **Fig 6B**). In contrast, the tissues in the wound areas treated with 6% ML exhibited a positive stain for collagen I (red fluorescence in **Fig 6D**). Semi-

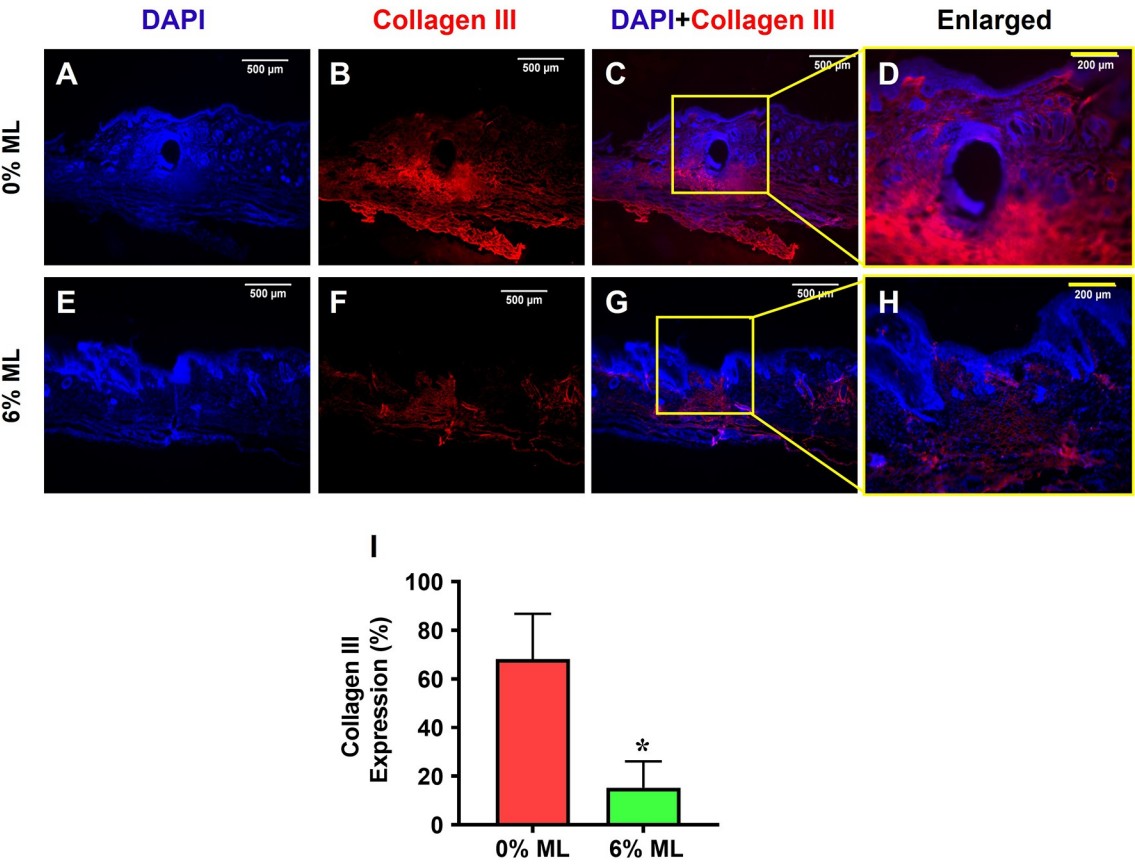

**Fig 7. Metformin lotion decreases collagen III levels in the wounded skin of rats after 10 days of wound healing.** The results show that 6% ML decreased collagen III expression in wounded skin areas (**E-H**). In contrast, wounds treated with 0% ML exhibited higher levels of collagen III expression compared to the 6% ML-treated wounds (**A-D**). Semi-quantification results are in agreement with these findings (**I**). *p < 0.001 compared to the 0% ML treated wound. ML: Metformin Lotion. Scale bars: 500 μm (white), 200 μm (yellow). Analysis was conducted using immunostaining.

quantitative results showed that over 55% of the cells in the wound areas treated with 0% ML expressed collagen III, whereas only about 19.8% of the cells in the areas treated with 6% ML showed similar expression (**Fig 6E**).

Immunostaining for collagen III expression corroborated the results from the Picro Sirius Red staining. It showed that over 68% of the cells in the wound areas of rats treated with 0% ML were positively stained for collagen III (**Fig 7A–7D, 7I**). In contrast, only 15.3% of the cells in the wound areas of rats treated with 6% ML demonstrated similar staining (**Fig 7E–7H and 7I**).

Additionally, immunostaining for α-SMA demonstrated that ML reduced α-SMA expression in the wounded skin areas (**Fig 8E–8H**). Higher levels of α-SMA were observed in the wound area treated with 0% ML (**Fig 8A–8D**). Semi-quantitative results showed that more than 77% of the cells in the wound areas of rats treated with 0% ML expressed α-SMA (**Fig 8I**), while only about 16.7% of the cells in the wound areas treated with 6% ML showed α-SMA expression (**Fig 8I**).

Further examination revealed that treatment with 6% ML significantly reduced TGF-β1 expression in the wound areas (**Fig 9E–9H**), with 70% of the cells in areas treated with 0% ML expressing TGF-β1 (**Fig 9A–9D and 9I**). In contrast, 11.2% of the cells in areas treated with 6% ML expressed TGF-β1 (**Fig 9E–9H and 9I**).

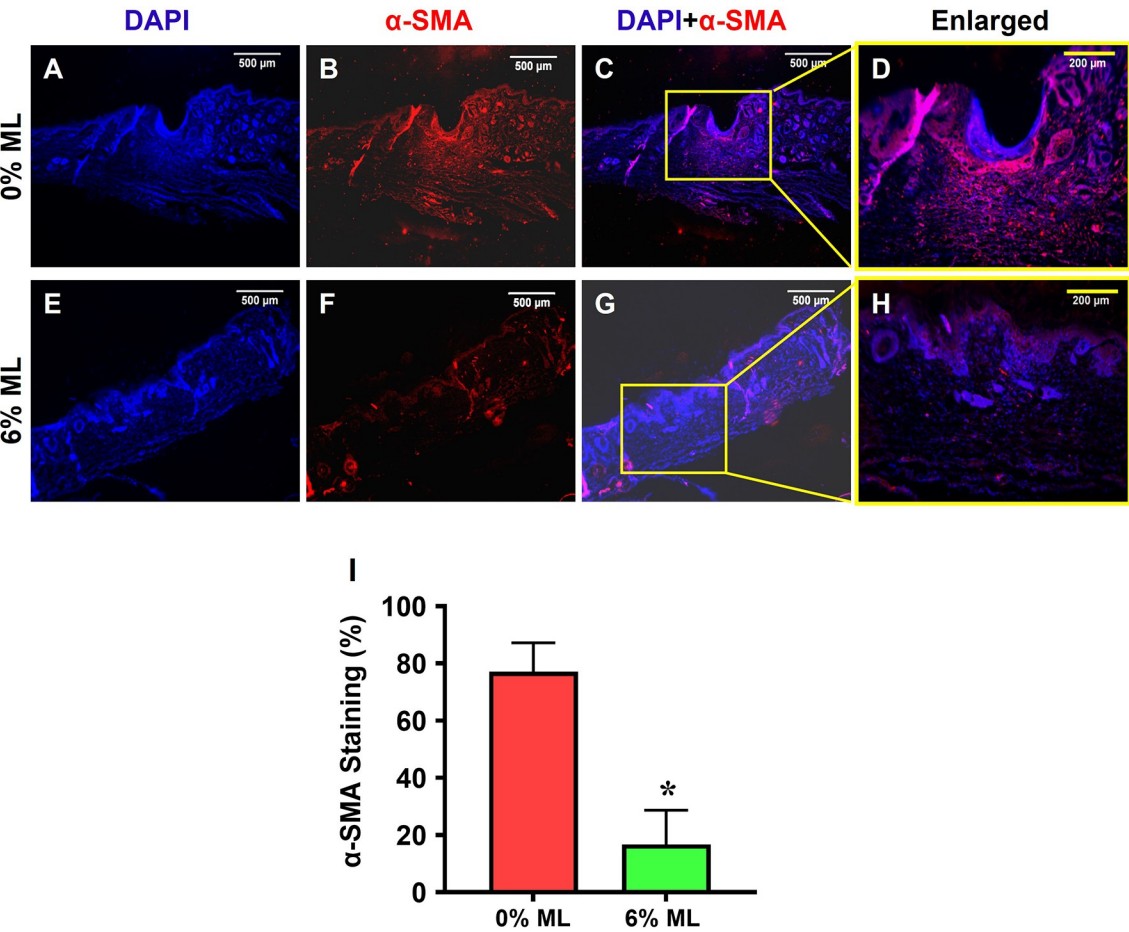

**Fig 8. Metformin lotion reduces α-SMA levels in the wounded skin of rats after 10 days of wound healing.** The results show that 6% ML decreases α-SMA expression in wounded skin areas (**E-H**). In contrast, 77% of the cells in the wounded skin areas treated with 0% ML express α-SMA (**A-D, I**), whereas about 16.7% of the cells in the wounded skin areas treated with 6% ML express α-SMA (**E-H, I**). *$p < 0.05$, compared to the 0% ML treated wounds. ML: Metformin Lotion. Scale bars: 500 μm (white), 200 μm (yellow). **C**: Merged images of **A** and **B**; **G**: Merged images of **E** and **F**; **D**: Enlarged yellow box area in **C**; **H**: Enlarged yellow box area in **G**. Analysis was conducted using immunostaining.

Moreover, immunostaining for phosphorylated AMPK (p-AMPK) indicated that 57% of the cells in the wound areas treated with 6% ML were positive for p-AMPK (**Fig 10D–10I**). Conversely, 5.1% of the cells in the wound areas treated with 0% ML showed p-AMPK expression (**Fig 10A–10D and 10I**).

Lastly, 51% of the cells in the wound areas that were treated with 6% ML expressed collagen I (**Fig 11E–11I**). In comparison, about 26% of the cells in the wound areas treated with 0% ML was positive for collagen I (**Fig 11A–11D and 11I**).

## Discussion

This study introduces a pioneering ML designed for topical administration to facilitate scarless healing of skin wounds. Our findings demonstrate a marked difference between wounds treated with the control lotion (0% ML) and those treated with a 6% ML. Notably, wounds treated with the 6% ML exhibited accelerated healing, resulting in the formation of tissue

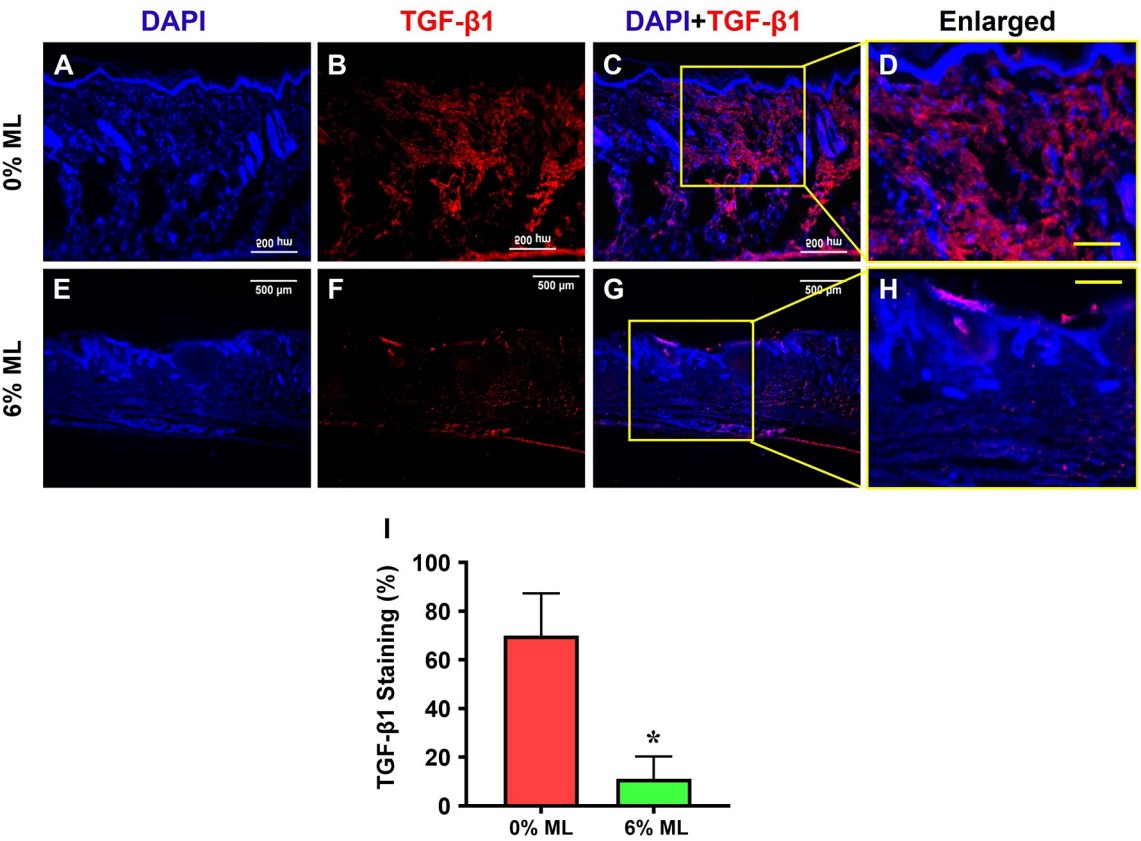

**Fig 9. Metformin lotion decreases TGF-β1 levels in the wounded skin of rats after 10 days of wound healing.** The results show that 6% ML inhibits TGF-β1 expression in wounded skin areas (**E-H**). In contrast, the wounds treated with 0% ML express higher levels of TGF-β1 (**A-D**) compared to 6% ML-treated wounds. Semi-quantification results agree with these findings (**I**). *p < 0.001, compared to the 0% ML treated wound. ML: Metformin Lotion. Scale bars: 500 μm (white), 200 μm (yellow). **C**: Merged images of **A** and **B**; **G**: Merged images of **E** and **F**; **D**: Enlarged yellow box area in **C**; **H**: Enlarged yellow box area in **G**. Analysis was conducted using immunostaining.

resembling normal skin, as opposed to the scar-like tissue formation observed with the control lotion treatment.

ELISA analysis revealed a significant anti-inflammatory effect associated with the 6% ML, as evidenced by decreased levels of inflammatory markers HMGB1 and IL-1β in the blood samples of rats treated with this formulation compared to those treated with the 0% ML. Immunostaining further supported these findings, showing that administration of the 6% ML inhibited the release of HMGB1 from cell nuclei into the skin tissue matrix. Additionally, the 6% ML demonstrated anti-fibrotic properties, as indicated by several key observations. Treatment with the 6% ML led to enhanced p-AMPK activity, decreased expression of TGF-β1, reduced numbers of α-SMA⁺ cells, decreased production of collagen III, and increased expression of collagen I. These findings suggest that 6% ML shows potential as a therapeutic agent for promoting scarless skin wound healing by reducing inflammation and fibrosis while enhancing tissue regeneration.

Recent studies have demonstrated that HMGB1 is liberated from cell nuclei into the wound area following tissue injury [12]. This released HMGB1 has the capability to attract inflammatory cells to the damaged tissues, thus initiating inflammation [31]. The involvement of HMGB1 in the progression of various inflammatory and autoimmune conditions such as rheumatoid arthritis, sepsis, cardiovascular disease, and cancer has been documented [32].

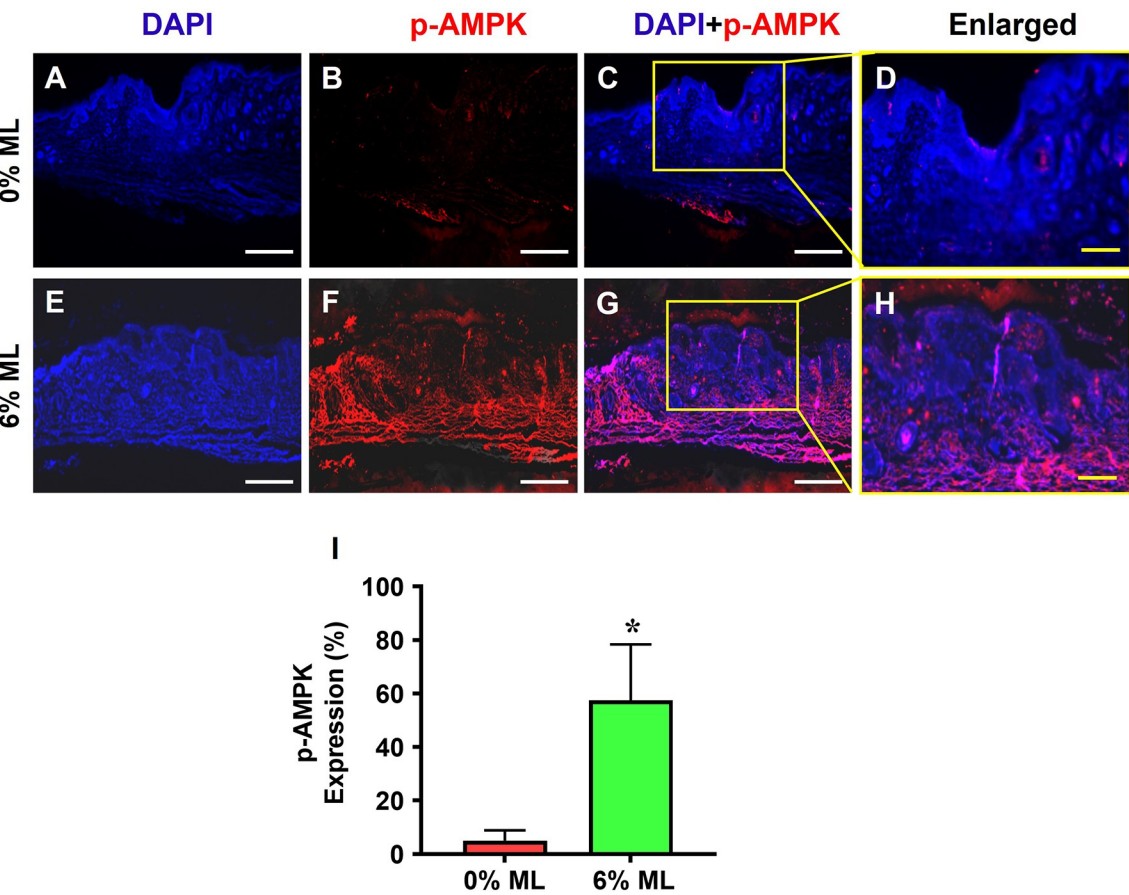

**Fig 10. Metformin lotion elevates p-AMPK in the wounded skin of rats after 10 days of wound healing.** The results show that 6% ML elevates p-AMPK expression in wounded skin areas (**E-H**). In contrast, the wounds treated with 0% ML express much lower levels of p-AMPK compared to the 6% ML-treated wounds (**A-D**). Semi-quantification results confirm these findings (**I**). *p < 0.001, compared to the 0% ML treated wound. ML: Metformin Lotion. Scale bars: 500 μm (white), 200 μm (yellow). **C:** Merged images of **A** and **B**; **G:** Merged images of **E** and **F**; **D:** Enlarged yellow box area in **C**; **H:** Enlarged yellow box area in **G**. Analysis was conducted using immunostaining.

Furthermore, our prior investigations into tendinopathy have revealed that HMGB1 release triggered by stress or injury can activate inflammatory pathways leading to tissue inflammation and degradation [33].

It has been reported that chronic inflammation can lead to fibrosis or scarring, characterized by the excessive accumulation of TGF-β1 [34] and collagen III [35]. Studies have highlighted the significant role of HMGB1, a potent mediator, in various fibrotic diseases such as liver fibrosis [26], renal fibrosis [15], and pulmonary fibrosis [16]. These investigations suggest that blocking HMGB1 release could mitigate inflammation and reduce scar tissue formation. Indeed, metformin, a specific inhibitor of HMGB1, has demonstrated considerable efficacy in reducing scar tissue formation and promoting wound healing. Our previous study has shown that metformin can inhibit HMGB1 release and facilitate tendon healing in wounded tissue [30]. Previous research has also indicated that metformin prevents peritendinous fibrosis by suppressing TGF-β1 signaling in a surgical rat model of flexor tendon peritendinous adhesion [36]. The findings of our current study align with these previous results.

Previous studies have revealed that metformin exerts its anti-inflammatory and anti-fibrotic effects through the activation of AMPK [37]. Clinical studies have demonstrated that patients

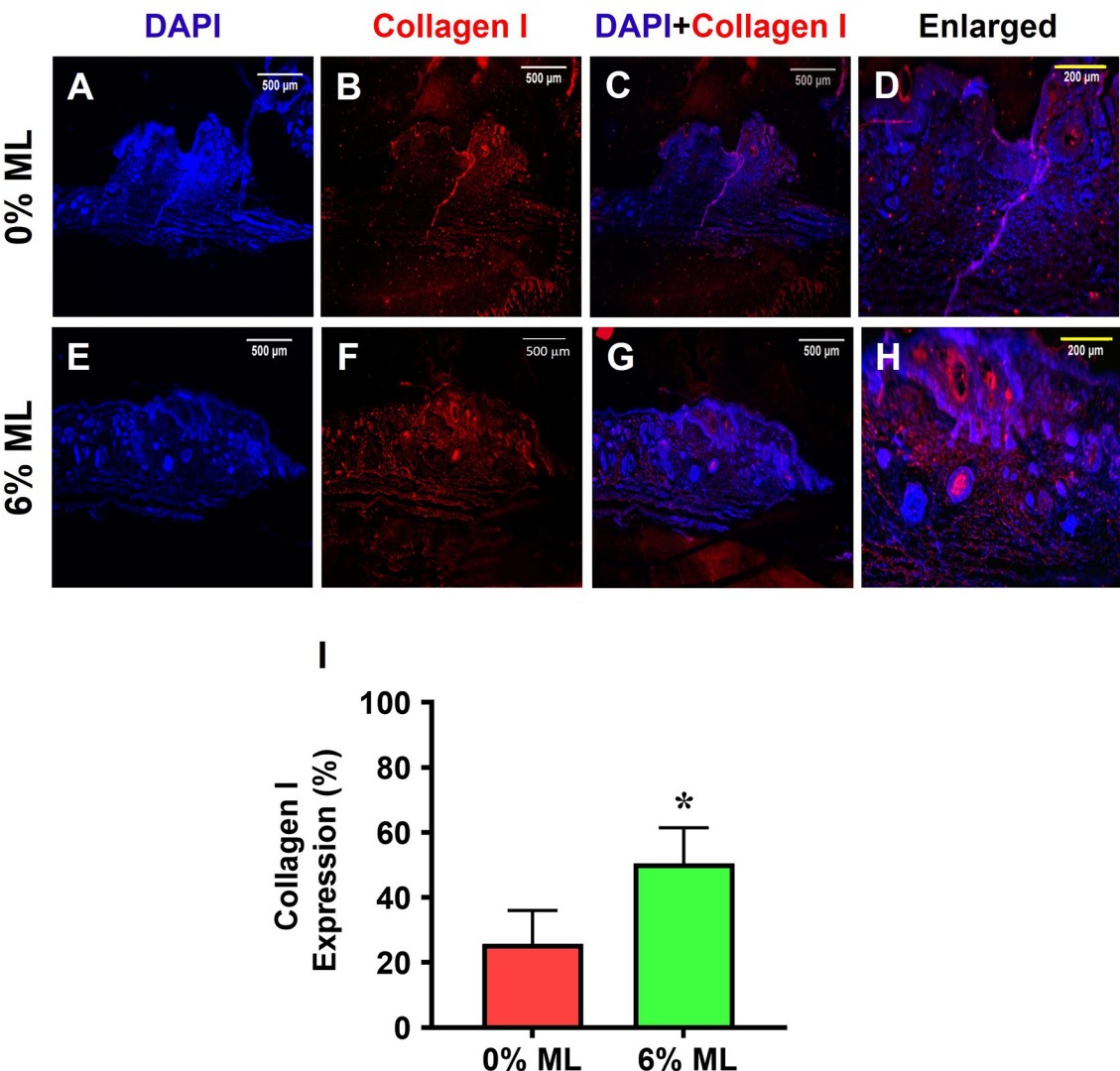

**Fig 11. Metformin lotion increases collagen I expression in the wounded skin of rats after 10 days of wound healing.** The results show that 6% ML increases collagen I expression in wounded skin areas (**E-H**). In contrast, wounds treated with 0% ML express much lower levels of collagen I compared to 6% ML-treated wounds (**A-D**). Semi-quantification results confirm these findings (**I**). **C**: Merged images of **A** and **B**; **G**: Merged images of **E** and **F**; **D**: Enlarged yellow box area in **C**; **H**: Enlarged yellow box area in **G**. *p < 0.001, compared to the 0% ML-treated wound. ML: Metformin Lotion. Scale bars: 500 μm (white), 200 μm (yellow). Analysis was conducted using immunostaining.

with pulmonary fibrosis exhibit reduced fibrotic activity and increased AMPK activation following treatment with metformin [25]. Additionally, animal research has indicated that metformin-induced AMPK activation facilitates the resolution of fibrosis in a manner dependent on AMPK in a mouse model of lung fibrosis [38]. Our results support these findings, as evidenced by elevated levels of pAMPK and reduced concentrations of TGF-β1 observed in the wounded skin area treated with 6% ML.

It is widely recognized that myofibroblasts, characterized by their abundant ECM production, play a pivotal role in the pathology of fibrotic disease [39]. α-SMA serves as a specific marker for myofibroblasts. Previous research has indicated that fibrotic scar tissue formation is promoted by anabolic metabolism in activated myofibroblasts [40]. The current study

demonstrates that 6% ML not only activates AMPK but also suppresses the recruitment of $\alpha$-SMA$^+$ myofibroblasts to the wound areas, thereby inhibiting scar tissue formation.

Promoting scarless wound healing remains a significant challenge for both clinicians and researchers. While metformin is already FDA-approved for oral use, its oral administration often entails adverse effects on the stomach, liver, and kidneys, and it tends to be less effective in treating skin wounds. Introducing metformin in the form of a lotion represents a novel approach that allows for direct delivery of the drug to the affected skin area, thereby enhancing wound healing.

The use of ML offers several advantages. By delivering metformin directly to the skin, it can achieve higher efficiency with smaller amounts of the drug compared to oral administration, while minimizing systemic side effects on internal organs. This study demonstrated that applying 6% ML to the wounded skin area promotes wound healing, reduces scar tissue formation, and facilitates the regeneration of normal, functional skin.

## Acknowledgments

We thank Bhavani P. Thampatty for assistance in the preparation of this manuscript.

## Author Contributions

**Conceptualization:** James H-C. Wang.

**Data curation:** Jianying Zhang, Derek Maloney.

**Formal analysis:** Jianying Zhang.

**Funding acquisition:** James H-C. Wang.

**Methodology:** Jianying Zhang, Kengo Shimozaki, Soichi Hattori, Vasyl Pastukh, Derek Maloney.

**Resources:** James H-C. Wang.

**Supervision:** James H-C. Wang.

**Validation:** Kengo Shimozaki, Soichi Hattori, Vasyl Pastukh.

**Writing – original draft:** Jianying Zhang.

**Writing – review & editing:** MaCalus V. Hogan, James H-C. Wang.

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
