## [Decision Letter · Decision Letter 0]

26 Aug 2024

PONE-D-24-30549Metformin Lotion Promotes Scarless Skin Tissue Formation through AMPK Activation, TGF-β1 Inhibition, and Reduced Myofibroblast NumbersPLOS ONE

Dear Dr. Wang,

Thank you for submitting your manuscript to PLOS ONE. After careful consideration, we feel that it has merit but does not fully meet PLOS ONE’s publication criteria as it currently stands. Therefore, we invite you to submit a revised version of the manuscript that addresses the points raised during the review process.  Please submit your revised manuscript by Oct 10 2024 11:59PM. If you will need more time than this to complete your revisions, please reply to this message or contact the journal office at plosone@plos.org. Please include the following items when submitting your revised manuscript:A rebuttal letter that responds to each point raised by the academic editor and reviewer(s). You should upload this letter as a separate file labeled 'Response to Reviewers'.A marked-up copy of your manuscript that highlights changes made to the original version. You should upload this as a separate file labeled 'Revised Manuscript with Track Changes'.An unmarked version of your revised paper without tracked changes. You should upload this as a separate file labeled 'Manuscript'.If applicable, we recommend that you deposit your laboratory protocols in protocols.io to enhance the reproducibility of your results. Protocols.io assigns your protocol its own identifier (DOI) so that it can be cited independently in the future. For instructions see: https://journals.plos.org/plosone/s/submission-guidelines#loc-laboratory-protocols. Additionally, PLOS ONE offers an option for publishing peer-reviewed Lab Protocol articles, which describe protocols hosted on protocols.io. Read more information on sharing protocols at https://plos.org/protocols?utm_medium=editorial-email&utm_source=authorletters&utm_campaign=protocols.

We look forward to receiving your revised manuscript.

Kind regards,

Naveed Ahmed, Ph.D

Academic Editor

PLOS ONE

Journal Requirements:

Reviewers' comments:

Reviewer's Responses to Questions

**Comments to the Author**

1. Is the manuscript technically sound, and do the data support the conclusions?

Reviewer #1: Yes

Reviewer #2: Yes

2. Has the statistical analysis been performed appropriately and rigorously? 

Reviewer #1: Yes

Reviewer #2: Yes

3. Have the authors made all data underlying the findings in their manuscript fully available?

Reviewer #1: Yes

Reviewer #2: Yes

4. Is the manuscript presented in an intelligible fashion and written in standard English?

Reviewer #1: Yes

Reviewer #2: Yes

5. Review Comments to the Author

Reviewer #1: Results are generally compelling and consistent. Specific comments are as follow:

1. Introduction

- Line 108. Mention the full term of TGF-β1 at its first mention

2. Material and method

- Line 140. Ten female Sprague Dawley (SD) rats, aged 3 months. Mention the average weight.

- Line 155. Mention the full term of IL-1β at its first mention

Reviewer #2: 19.08.2024

In this manuscript, the authors described "Metformin Lotion Promotes Scarless Skin Tissue Formation through AMPK Activation, TGF-β1 Inhibition, and Reduced Myofibroblast Numbers". The study will be beneficial for the literature. The report is an interesting study, but it needed some suggestions for publication.

Here are the concerns for the authors;

Generally;

1. The affiliations should be correctly written.

2. More information about metformin should be given in the introduction.

3. What is the purpose of choosing Metformin lotion? You should indicate in the article why you chose a lotion and not a cream, ointment or gel.

4. 6% ML was used according to what? Have any experiments been carried out to determine this concentration? If so, the results and explanations should be given.

5. It would be better if photographs could be added to be more descriptive about the wound study applied in rats.

6. “A 2 cm incised wound was carefully created on the skin over each Achilles tendon area using a scalpel, after which the wounds were sutured.”, no anesthesia was administered at this stage? It should be indicated how the anesthesia was administered and how much was given to rats.

7. G power analysis related to the animals used should be given.

8. Why was a healthy control group or a positive control group not included in the groups? Group 1 is a negative control in which a wound was created and given 0% ML. Therefore, would it not have been correct to compare it with a group without wound formation (healthy control) or with a lotion used routinely (positive control)?

9. “Achilles tendon area using a scalpel, after which the wounds were sutured.”, and what was done to prevent any contamination afterwards?

10. Fluorescent figures should be given sharper.

11. “Semi-quantification results indicate more than 46% of the cells around the wound area of the rat skin treated with 6% ML kept HMGB1 in the nuclei; however, less than 3.7% of the cells in the wound area of the rat skin treated with 0% ML showed HMGB1 in the nuclei”, figure legends contain bulleted sentences. These sentences should be included in the results section, not in the figure legends. Review the legends of each figure.

6. PLOS authors have the option to publish the peer review history of their article (what does this mean?). If published, this will include your full peer review and any attached files.

Reviewer #1: **Yes: **Hany M Fayed

Reviewer #2: No

---

## [Author Response · Author response to Decision Letter 0]

5 Sep 2024

Reviewer #1: Results are generally compelling and consistent. Specific comments are as follow:

1. Introduction

- Line 108. Mention the full term of TGF-β1 at its first mention

2. Material and method

- Line 140. Ten female Sprague Dawley (SD) rats, aged 3 months. Mention the average weight.

- Line 155. Mention the full term of IL-1β at its first mention

Response: Thank you. We have revised the relevant text in accordance to your suggestions.

Reviewer #2: In this manuscript, the authors described "Metformin Lotion Promotes Scarless Skin Tissue Formation through AMPK Activation, TGF-β1 Inhibition, and Reduced Myofibroblast Numbers". The study will be beneficial for the literature. The report is an interesting study, but it needed some suggestions for publication. Here are the concerns for the authors.

1. The affiliations should be correctly written.

Response: We have done so.

2. More information about metformin should be given in the introduction.

Response: More information about metformin has been given now.

3. What is the purpose of choosing Metformin lotion? You should indicate in the article why you chose a lotion and not a cream, ointment or gel.

Response: Met lotion is a water-based product with a higher concentration of metformin, lower viscosity, and better skin penetration compared to creams, ointments, and gels, as metformin is a water-soluble compound. 

4. 6% ML was used according to what? Have any experiments been carried out to determine this concentration? If so, the results and explanations should be given.

Response: To determine the appropriate dose of metformin lotion, we conducted a skin permeation test using the Franz Cell system. We found that the 6% Met-lotion effectively permeated both pig and human skin. Additionally, a pharmacokinetic study in mice demonstrated that the 6% Met-lotion resulted in more than threefold higher concentrations of metformin in serum and the Achilles tendon compared to oral administration of the same dose. Based on these findings, we selected the 6% Met-lotion for use in the rat skin wound healing study.

5. It would be better if photographs could be added to be more descriptive about the wound study applied in rats. 

Response: The pictures have been added to describe the skin wound healing model (please see Fig. 1).

6. “A 2 cm incised wound was carefully created on the skin over each Achilles tendon area using a scalpel, after which the wounds were sutured”, no anesthesia was administered at this stage? It should be indicated how the anesthesia was administered and how much was given to rats.

Response: Thank you. This is an important question. We did perform anesthesia during our surgical procedures. See tracked changes in the revision. 

7. G-power analysis related to the animals used should be given.

Response: Our study consists of two groups: the vehicle control group and the 6% ML group. We estimated the effect size of the Met treatment to be a minimum of 1.8 (Zhang et al., Pharmaceuticals 2023, 16, 1739).

The following parameters were used to calculate the required sample size using G-Power Software.

• Effect size = 1.8

• α = 0.05

• Power (1 - β) = 0.8

• Allocation ratio (N2/N1) = 1

The resulting sample size is 10, with 5 mice in each of the two groups.

8. Why was a healthy control group or a positive control group not included in the groups? Group 1 is a negative control in which a wound was created and given 0% ML. Therefore, would it not have been correct to compare it with a group without wound formation (healthy control) or with a lotion used routinely (positive control)?

Response: We used the intact skin adjacent to the wound area on the same rat as a healthy control. Additionally, we compared the healing results with the intact skin from healthy, untreated rats.

9. “Achilles tendon area using a scalpel, after which the wounds were sutured,” and what was done to prevent any contamination afterwards?

Response: The surgery was performed under sterile conditions. The surgical area was sprayed with 70% ethanol and swabbed with iodine prior to the procedure. The animals received carprofen at a dose of 5 mg/kg during surgery and twice daily for 4 days post-surgery. These procedures were implemented to prevent contamination.

10. Fluorescent figures should be given sharper.

Response: The original figures are of high quality, but the problem arises when the file is converted to PDF format, which reduces the image quality of all figures. You may download the original figure images we submitted and review them instead. 

11. “Semi-quantification results indicate more than 46% of the cells around the wound area of the rat skin treated with 6% ML kept HMGB1 in the nuclei; however, less than 3.7% of the cells in the wound area of the rat skin treated with 0% ML showed HMGB1 in the nuclei”, figure legends contain bulleted sentences. These sentences should be included in the results section, not in the figure legends. Review the legends of each figure.

Response: We have revised all the legends accordingly. Thank you.

---

## [Editor Report · Decision Letter 1]

16 Sep 2024

Metformin Lotion Promotes Scarless Skin Tissue Formation through AMPK Activation, TGF-β1 Inhibition, and Reduced Myofibroblast Numbers

PONE-D-24-30549R1

Dear Dr. Wang,

We’re pleased to inform you that your manuscript has been judged scientifically suitable for publication and will be formally accepted for publication once it meets all outstanding technical requirements.

Kind regards,

Naveed Ahmed, Ph.D

Academic Editor

PLOS ONE
---

## [Editor Report · Acceptance letter]

18 Sep 2024

PONE-D-24-30549R1 

PLOS ONE

Dear Dr. Wang, 

I'm pleased to inform you that your manuscript has been deemed suitable for publication in PLOS ONE. Congratulations! Your manuscript is now being handed over to our production team.

Kind regards, 

on behalf of

Dr. Naveed Ahmed 

Academic Editor

PLOS ONE